# Adjustable Quantile-Guided Diffusion Policy for Diverse Behavior Generation in Offline RL

## Abstract

Offline Reinforcement Learning (RL) addresses the challenge of learning optimal policies from pre-collected data, making it a promising approach for real-world applications where online interactions with an environment are costly or impractical. We propose an offline RL method named Quantile-Guided Diffusion Policy (qGDP), which trains a quantile network to label the training dataset and uses these labeled samples to train the diffusion model and generate new samples with the trained model according to classifier-free guidance. qGDP can adjust the preference of sample generation between imitating and improving behavioral policies by adjusting the input condition and changing the guidance scale without re-training the model, which will significantly reduce the cost of tuning the algorithm. qGDP exhibits exceptional generalization capabilities and allows easy adjustment of action generation preferences without model retraining, reducing computational costs. Experimental results on the D4RL dataset demonstrate state-of-the-art performance and computational efficiency compared to other diffusion-based methods.

## 1 Introduction

Offline reinforcement learning (RL), also known as batch RL, aims to learn the effective policies from a given training dataset without real-time interaction with the environment (Levine et al., 2020; Prudencio et al., 2023). This paradigm proves highly beneficial in situations where conducting interactions online is not feasible, like robotics (Singh et al., 2022), education (Singla et al., 2021), healthcare (Liu et al., 2020), and autonomous driving (Kiran et al., 2021). However, the inability to interact with the environment online also makes offline RL difficult. The training dataset is limited and cannot effectively cover the entire possible state-action space, so agents are unable to estimate the value of the out-of-distribution actions accurately. When these actions are overestimated, the learned policy would be poor.

The most intuitive and common approach is to constrain the discrepancy between the learned policy and the behavioral policy. Some studies (Fujimoto et al., 2019; Kumar et al., 2019; Fujimoto & Gu, 2021; Wu et al., 2019) introduce measures of distributional differences, such as KL divergence, MMD, and Wasserstein distance, as regularization terms during the policy evaluation or improvement step of actor-critic methods (Prudencio et al., 2023). These regularization terms serve to balance the improvement and imitation of the behavioral policy. These methods have achieved some success, but the performance on some complex tasks or datasets still needs to be improved.

Wang et al. (2022) states that policies represented as Gaussian distributions struggle to accurately capture the multi-modal, skewed, and interdependent nature of behavior policies. They also introduce Diffusion Q-Learning (DQL), which utilizes a diffusion model (Ho et al., 2020; Sohl-Dickstein et al., 2015; Song & Ermon, 2019) to generate actions and employs two loss functions to encourage the diffusion to match the training dataset's action distribution and guide it towards high-value actions. Diffuser proposed by Janner et al. (2022) is employed for solving offline RL problems. A diffusion model is trained to imitate the training dataset, while during sampling, the derivative of the cumulative reward with respect to noised trajectories is used to guide the generation of high-value trajectories. The introduction of diffusion enhances the expressiveness of the policy, but the guidance of value or reward functions during the training or sampling process involves extensive gradient computations, significantly increasing computational costs. Implicit DQL (IDQL) proposed by

Hansen-Estruch et al. (2023) re-weights samples in the training dataset based on action values and utilizes diffusion to mimic the weighted action distribution. It can not introduce additional gradient computations during the training and sampling process, but it potentially constrains the performance of the policy.

Addressing the problems of previous diffusion-based offline RL methods, we present a novel offline RL method named Quantile-Guided Diffusion Policy (qGDP), which leverages a quantile network to label dataset samples and trains a diffusion model conditioned the given state and quantile. qGDP trains a quantile network to provide different quantiles of the action value distribution under the behavioral policy for each state and labels the samples based on the relationship between action values and different quantiles. Then the diffusion model is trained like the algorithm of classifier-free guided diffusion, which replaces sampling in the direction of the gradient of the classifier with the direction of the difference between the conditional and unconditional models. In this way, the training and sampling processes of qGDP do not involve gradients with respect to action value functions, leading to lower computational costs. Moreover, qGDP exhibits a remarkable generalization capability, as the conditional diffusion demonstrates in text-to-image generation tasks, where it can be used to generate actions beyond the quantile of $100\%$. Most importantly, qGDP can alter its action generation preferences by straightforwardly adjusting hyperparameters during the sampling process without retaining the model, as is required by other offline RL methods. This further mitigates the overall computational cost associated with the entire training process, including hyperparameter tuning.

We compared qGDP with other diffusion-based offline reinforcement learning methods, including Diffuser (Janner et al., 2022), DQL (Wang et al., 2022), and IDQL (Hansen-Estruch et al., 2023), in several simple bandit tasks. qGDP-Q and qGDP-GeQ generate samples proportional to the quantile and maintain high diversity. When the input quantile is greater than $100\%$, the learned model can generate higher-value samples based on the reward function's trends. The experimental results on the D4RL dataset demonstrate that our approach has achieved state-of-the-art performance. In terms of runtime comparison, qGDP demonstrates computational efficiency, requiring fewer iterations and shorter runtimes compared to DQL.

## 2 RELATED WORK

Offline RL requires balancing between maximizing cumulative rewards and maintaining similarity with the behavioral policy. Excessive deviation from the behavioral policy introduces extrapolation error into the dynamic programming process, potentially leading to overestimation of value and performance degradation. (Fujimoto et al., 2019) BCQ (Fujimoto et al., 2019) uses CVAE (Sohn et al., 2015) to imitate behavioral policies and uses DDPG to optimize perturbations to the CVAE model. TD3+BC (Fujimoto & Gu, 2021) improves the policy using TD3 (Fujimoto et al., 2018) while employing Maximum Likelihood Estimation (MLE) to constrain the policy. BRAC (Wu et al., 2019) attempted various constrained methods such as KL divergence, kernel MMD and Wasserstein distance. AWR (Peng et al., 2019), AWAC (Nair et al., 2020) and CRR (Wang et al., 2020) constraint the policy by regress it onto weighted target actions. In addition, some work avoids policy deviation from the behavioral policy by reducing the value estimation of actions outside the training dataset (Kumar et al., 2020; Kostrikov et al., 2021).

Wang et al. (2022) indicates that unsuitable regular terms and Gaussian Policy Classes limit the expressiveness of policies and proposes DQL, which introduces a diffusion model to imitate behavioral policies and guide the generation process of diffusion model through action-value networks. IDQL (Hansen-Estruch et al., 2023) used the diffusion model to fit the distribution of actions weighted based on action values. Diffuser (Janner et al., 2022) utilizes diffusion to mimic trajectories in the training dataset and employs a reward function to guide the generation of trajectories. SfBC (Chen et al., 2022) employs diffusion to model behavioral policies and utilizes diffusion for in-sample planning and action value evaluation. IDQL and SfBC would not generate unseen actions, but IDQL and Diffuser could generate unseen actions guided by the value function. When the value function changes smoothly, the actual value of these unseen actions is likely to be higher than the actions in the training dataset. Our approach also uses diffusion to play the role of a policy function, but we use the algorithm similar to classifier-free guidance to train a diffusion model with quantile as guidance. The computational cost of the training and sampling process in this approach is

lower compared to DQL and Diffuser. Additionally, it can be controlled whether diffusion generates unseen actions with potential high value by adjusting hyperparameters during the sampling process.

## 3 PRELIMINARIES

### 3.1 OFFLINE REINFORCEMENT LEARNING

Reinforcement Learning (RL) aims to solve sequential decision problems that are formally defined as Markov Decision Processes (MDP). An MDP is defined as a tuple $(\mathcal{S}, \mathcal{A}, P, R, \gamma, d_0)$ where $\mathcal{S}$ denotes the state space, $\mathcal{A}$ denotes the action space, $P(\mathbf{s}'|\mathbf{s}, \mathbf{a}) : \mathcal{S} \times \mathcal{A} \times \mathcal{S} \to [0, 1]$ denotes the state transfer probability function, $R(\mathbf{s}, \mathbf{a}) : \mathcal{S} \times \mathcal{A} \to \mathbb{R}$ denotes the reward function, $\gamma$ denotes the discount factor, and $d_0$ denotes the distribution of initial states (Sutton & Barto, 2018). The purpose of RL is to learn a policy $\pi$ to maximize the expected cumulative discount rewards: $J(\pi) = \mathbb{E}_{\tau \sim p_\pi(\tau)}[\sum_{t=0}^{\infty} \gamma^t R(\mathbf{s}_t, \mathbf{a}_t)]$.

In the offline RL setting (Fu et al., 2020), the agent is not allowed to interact with the environment but has access to a static dataset $\mathcal{D} = \{(\mathbf{s}, \mathbf{a}, r, \mathbf{s}')\}$ collected by a behavioral policy $\mu(\mathbf{a}|\mathbf{s})$. Agents are required to utilize this dataset to find a policy $\pi(\mathbf{a}|\mathbf{s})$ that performs as well as possible in the original environment.

### 3.2 DIFFUSION MODEL

Diffusion models (Ho et al., 2020; Sohl-Dickstein et al., 2015; Song & Ermon, 2019) focus on modeling how to transform, step by step, from a simple distribution $p(\mathbf{x}_T)$ (e.g., a Gaussian distribution $\mathcal{N}(\mathbf{x}_T; \mathbf{0}, \mathbf{I})$) into a complex distribution of the interested data $p(\mathbf{x}_0)$. The forward diffusion chain is obtained by adding noise to the data $\mathbf{x}_0 \sim p(\mathbf{x}_0)$ step by step:

$$q(\mathbf{x}_{1:T}|\mathbf{x}_0) := \prod_{t=1}^{T} q(\mathbf{x}_t|\mathbf{x}_{t-1}), \quad q(\mathbf{x}_t|\mathbf{x}_{t-1}) := \mathcal{N}(\sqrt{1-\beta_t}\mathbf{x}_{t-1}, \beta_t \mathbf{I}), \tag{1}$$

where $t \in (0, T]$ is , and $(\beta_1, \ldots, \beta_T)$ is a pre-defined schedule. The reverse diffusion chain is

$$p_\theta(\mathbf{x}_{0:T}) := \mathcal{N}(\mathbf{x}_T; \mathbf{0}, \mathbf{I}) \prod_{t=1}^{T} p_\theta(\mathbf{x}_{t-1}|\mathbf{x}_t), \tag{2}$$

where $p_\theta(\mathbf{x}_{t-1}|\mathbf{x}_t)$ is the model for approximating the true posterior and is optimized by maximizing the variational lower bound, $\mathbb{E}_q[\ln(\frac{p_\theta(\mathbf{x}_{0:T})}{q(\mathbf{x}_{1:T}|\mathbf{x}_0)})]$ (Jordan et al., 1999; Blei et al., 2017).

Following DDPM (Ho et al., 2020), the model $p_\theta(\mathbf{x}_{t-1}|\mathbf{x}_t)$ is defined as

$$\mathcal{N}(\mathbf{x}_{t-1}; \boldsymbol{\mu}_\theta(\mathbf{x}_t, t), \boldsymbol{\Sigma}_\theta(\mathbf{x}_t, t)) = \mathcal{N}(\mathbf{x}_{t-1}; \frac{1}{\sqrt{\alpha_t}}(\mathbf{x}_t - \frac{\beta_t}{\sqrt{1-\bar{\alpha}_t}}\boldsymbol{\epsilon}_\theta(\mathbf{x}_t, t)), \beta_t \mathbf{I}), \tag{3}$$

where $\alpha_t := 1 - \beta_t$, and $\bar{\alpha}_t = \prod_{i=1}^{t} \alpha_i$. And the lower bound on $p_\theta(\mathbf{x}_{t-1}|\mathbf{x}_t)$ can be optimized by using the simplified objective:

$$\mathbb{E}_{t \sim [1,T], \boldsymbol{\epsilon} \sim \mathcal{N}(\mathbf{0}, \mathbf{I}), \mathbf{x}_0 \sim p(\mathbf{x}_0)} \|\boldsymbol{\epsilon} - \boldsymbol{\epsilon}_\theta(\sqrt{\bar{\alpha}_t}\mathbf{x}_0 + \sqrt{1-\bar{\alpha}_t}\boldsymbol{\epsilon}, t)\|^2. \tag{4}$$

To improve the quantity of the generated samples, Dhariwal & Nichol (2021) introduces gradients from the classifier into the generation process, and the reverse process is replaced with:

$$p_\theta(\mathbf{x}_{t-1}|\mathbf{x}_t) = \mathcal{N}(\boldsymbol{\mu}_\theta(\mathbf{x}_t|y) + \lambda \boldsymbol{\Sigma}_\theta(x_t|y) \nabla_{\mathbf{x}_t} \log p_\phi(y|\mathbf{x}_t), \boldsymbol{\Sigma}_\theta(\mathbf{x}_t|y)), \tag{5}$$

where $y$ is the target class, $\lambda$ is the guidance scale and $\phi(y|\mathbf{x}_t)$ denotes the probability of the class being predicted to be $y$ by the classifier.

To avoid the noise introduced by the classifier training process, Ho & Salimans (2021) proposes classifier-free guidance which does not require an additional classifier model. For this method, the diffusion model is $\boldsymbol{\epsilon}_\theta(\mathbf{x}_t|y)$, where the label $y$ can be the class or a null label $\emptyset$. During training, the label of the data would be replaced with $\emptyset$ with a small probability. When the label is replaced, the unconditional model is trained. During sampling, the output of the diffusion model is set to:

$$\hat{\boldsymbol{\epsilon}}_\theta(\mathbf{x}_t|y) = \boldsymbol{\epsilon}_\theta(\mathbf{x}_t|\emptyset) + \lambda(\boldsymbol{\epsilon}_\theta(\mathbf{x}_t|y) - \boldsymbol{\epsilon}_\theta(\mathbf{x}_t|\emptyset)) \tag{6}$$

The guidance scale $\lambda$ tunes the magnitude of the output away from $\boldsymbol{\epsilon}_\theta(\mathbf{x}_t|\emptyset)$ along the direction of $(\boldsymbol{\epsilon}_\theta(\mathbf{x}_t|y) - \boldsymbol{\epsilon}_\theta(\mathbf{x}_t|\emptyset))$.

## 4 QUANTILE-GUIDED DIFFUSION POLICY

In this section, we propose a quantile-guided diffusion-based policy named Quantile-Guided Diffusion Policy (qGDP). First, we describe how to train a quantile network and employ it to label the samples in the given dataset. Then, we train a diffusion model conditioned on state and label using these labeled samples based on the algorithm of training classifier-free guidance diffusion. Finally, the main advantages of the algorithm are illustrated: i) the model training and sampling are fast; ii) The model does not require to be retrained when we regulate the target policy to prefer to imitate the behavioral policy or to generate the actions with a higher value estimated by the critic.

In order to enable diffusion to generate high-quality actions based on the current environmental state, it is necessary to annotate the samples in the dataset with labels that are relevant to action values. An action-value prediction network denoted as $Q_{\hat{\phi}}(s, a)$, is trained using Implicit Q-Learning (IQL, (Kostrikov et al., 2021)). The loss calculation process for this network is as follows:

$$\mathcal{L}_V(\varphi) = \mathbb{E}_{(s,a)\sim\mathcal{D}} L_2^{\tau_c}\big(Q_{\hat{\phi}}(s, a) - V_\varphi(s)\big) \tag{7}$$

$$\text{where} \quad L_2^{\tau_c}(u) = |\tau - \mathbf{1}_{u<0}|u^2 \tag{8}$$

$$\mathcal{L}_Q(\phi) = \mathbb{E}_{(s,a,s')\sim\mathcal{D}}\big(r(s, a) + V_\varphi(s') - Q_\phi(s, a)\big)^2 \tag{9}$$

where $V(\varphi)$ denotes the state value network, $\tau$ is a pre-defined parameter that governs the $\tau$-quantile evaluation of the value of state $s$ under the action value distribution $Q(s, a), a \sim \pi_b(s)$ with respect to the behavioral policy $\pi_b(s)$.

Based on the action value network $Q_{\hat{\phi}}(s, a)$, we additionally train a quantile network to provide different quantiles of the action value distribution corresponding to the behavioral policy under state $s$, with the loss function defined as follows:

$$\mathcal{L}_{QR}(\eta) = \sum_{i=1}^{N} \mathbb{E}_{(s,a)\sim\mathcal{D}} L^{\tau_i}\big(Q_{\hat{\phi}}(s, a) - V_\eta^i(s)\big) \tag{10}$$

$$\text{where} \quad L^{\tau_i}(u) = u(\tau_i - \mathbf{1}_{u<0}) \tag{11}$$

$N$ represents the number of selected quantiles, and $\tau_i{}_{i=1}^N$ denotes the values of these quantiles. The first and last quantiles are set to 0.0001 and 0.9999, respectively, while the remaining quantiles are defined as $\tau_i = \frac{i}{N-1}$.

Using the quantile network $V_\eta^i(s)i = 1^N$ and the action value network $Q_{\hat{\phi}}(s, a)$ as the foundation, we can assign labels to samples $(s, a)$ in the dataset that are correlated with the relative magnitude of action values. We establish two labeling methods. One directly based on the quantile interval corresponding to the action value of the sample, and the label $y$ is set to

$$\sum_i \mathbf{1}_{Q_{\hat{\phi}}(s,a)>V_\eta^i(s)}. \tag{12}$$

The diffusion model trained using this labeling method is denoted as qGDP-Quantile, abbreviated as qGDP-Q. The other method is based on whether the action value exceeds a certain quantile, allowing each sample to have multiple labels, with one label randomly selected during training. The label set is

$$\{i|Q_{\hat{\phi}}(s, a) > V_\eta^i(s)\}. \tag{13}$$

Models trained using this labeling method are denoted as qGDP-GreaterThanQuantile, abbreviated as qGDP-GeQ.

In the training process, we use the training strategy of classifier-free guided diffusion. For the labeled sample $(s, a, y)$, the loss of the diffusion model is

$$\mathbb{E}_{t\sim[1,T],\epsilon\sim\mathcal{N}(0,I),(s,a,y)\sim\mathcal{D}}\|\epsilon - \epsilon_\theta(\sqrt{\bar{\alpha}_t}a + \sqrt{1 - \bar{\alpha}_t}\epsilon, s, y, t)\|^2. \tag{14}$$

In the sampling process, the conditional diffusion model utilizes the given label $y$ and state $s$ to generate action $a$:

$$\mathcal{N}(a^{i-1}; \mu_\theta(a^i, s, y, i), \Sigma_\theta(a^i, s, y, i)) = \mathcal{N}(a^{i-1}; \frac{1}{\sqrt{\alpha_i}}(a^i - \frac{\beta_t}{\sqrt{1 - \bar{\alpha}_t}}\hat{\epsilon}_\theta(a^i, s, y, i)), \beta_i I), \tag{15}$$

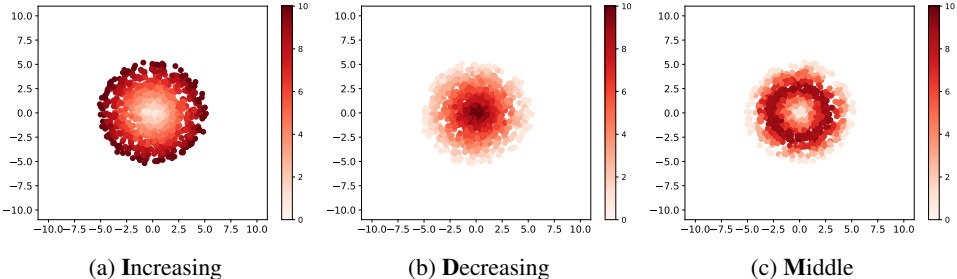

(a) **I**ncreasing         (b) **D**ecreasing         (c) **M**iddle

Figure 1: Three toy datasets with the same sample distribution but different reward functions. In the first dataset 1a, the rewards of samples increase with the distances from the center increasing. In the second dataset 1b, rewards decrease as the distance increases. In the third dataset 1c, the rewards are maximized for intermediate distances, whereas rewards are relatively smaller for distances either too far from or too close to the center. The color indicates the value of the reward.

$$\hat{\epsilon}_\theta(\boldsymbol{a}^i|\boldsymbol{s}, y) = \epsilon_\theta(\boldsymbol{a}^i|\boldsymbol{s}, \emptyset) + \lambda(\epsilon_\theta(\boldsymbol{a}^i|\boldsymbol{s}, y) - \epsilon_\theta(\boldsymbol{a}^i|\boldsymbol{s}, \emptyset)) \tag{16}$$

Due to the inherent uncertainties in value assessment, relying solely on the maximum quantile as a condition does not always yield optimal actions. This model allows for the adjustment of the generated action distribution and, consequently, the modification of behavioral patterns based on input quantiles denoted as $y$ or guidance weights represented by $\lambda$. In other words, **a) this model can flexibly modify its preferences between imitative behavioral strategies and the generation of high-value actions without requiring retraining**. Furthermore, **b) the training and sampling processes of this model do not involve gradients with respect to action value functions, resulting in lower computational costs**. Additionally, conditional diffusion exhibits a generalization capability in text-to-image generation tasks. We can **c) explore the use of this model to generate actions beyond the quantile of** $100\%$. Subsequent experiments confirm the presence of this generalization ability in the trained model.

## 5    THE DISTRIBUTIONS OF GENERATED SAMPLES WITH DIFFERENT GUIDANCE METHODS

In this section, we will employ several rudimentary datasets to validate distinctions among policies learned through various diffusion-based offline reinforcement learning methodologies.

**Datasets**    We construct three simple bandit tasks and their corresponding datasets. All three datasets contain $10,000$ samples and have the same sample distribution, but different reward functions. The rewards for the samples in the first dataset increase roughly with distance from the center. The rewards in the second dataset decrease with distance. The rewards in the third dataset are largest for medium distances and smaller for either too large or too small a distance from the center. The samples and corresponding rewards in these data sets are shown in Figure 1.

We extracted the process of using reward functions to guide the diffusion model in generating high-value actions from Diffuser, DQL, and IDQL, and compared these generation processes with our proposed method. We selected hyperparameters from each algorithm that control imitation and enhancement of behavioral policies and adjusted these hyperparameters to generate different distributions. The overall results are shown in Figure 2

By observing the generated results of these methods, we can see that each approach can generally approximate the distribution in the dataset or maximize the value of generated samples by adjusting hyperparameters. However, there are noticeable differences in the distribution of generated samples and their performance on specific datasets.

Diffuser is capable of generating diverse samples, and the value of samples generated on the Increasing and Decreasing datasets is positively correlated with the guidance scale. However, the highest-value samples on the Middle dataset do not correspond to the largest guidance scale. One possible reason for this is that a larger guidance scale causes the model's output to advance too

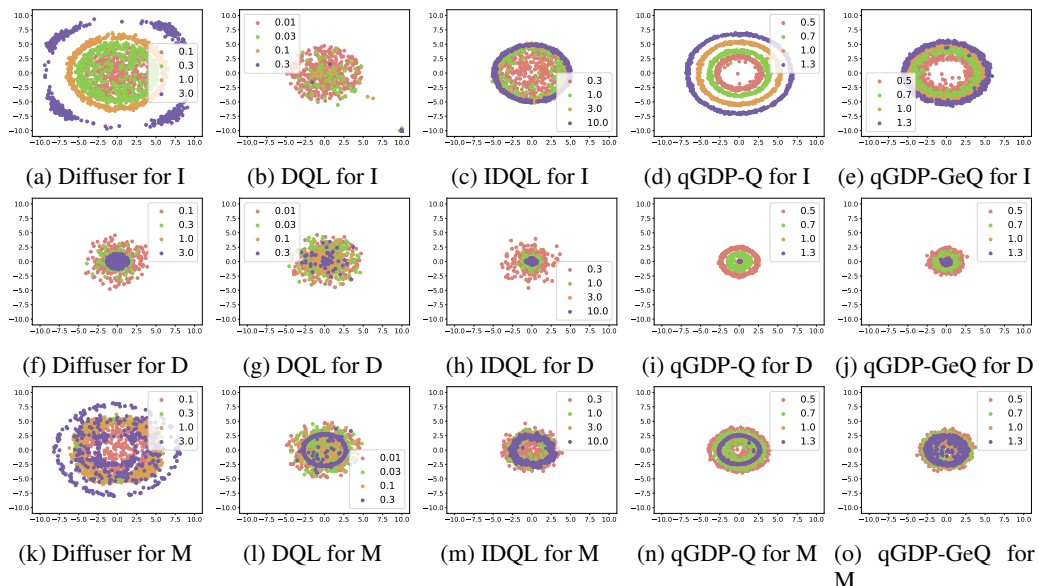

Figure 2: The results of applying different methods to three toy datasets. Rows one to three correspond to the datasets of Increasing, Decreasing, and Middle, respectively. The columns correspond to the methods of Diffuser, DQL, IDQL, qGDP-Q, and qGDP-GeQ, respectively. Samples of different colors correspond to different hyperparameters. Overall, these methods can balance between imitating the original data and maximizing the reward by adjusting hyperparameters. However, qGDP-Q and qGDP-GeQ exhibit a closer relationship between the samples generated under different hyperparameters (quantiles) and the rewards, with a more concentrated sample distribution.

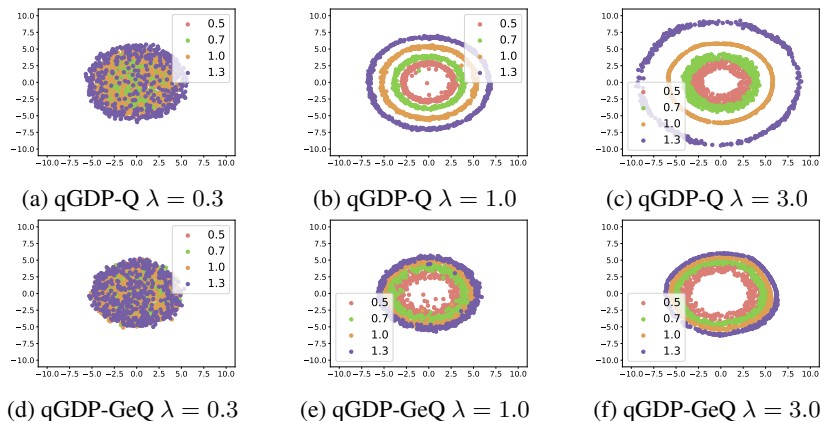

Figure 3: Under different guidance weights, the samples generated by qGDP-Q and qGDP-GeQ when applied to the dataset of Increasing. As the scale increases, generated samples tend to diverge from the original sample distribution and encompass regions associated with higher rewards; conversely, as the scale decreases, generated samples tend to converge towards the original sample distribution and cover a broader range of reward-associated regions.

far along the gradient of the value function. The value of generating samples with DQL at a high guidance scale is mostly high, but it still produces a significant number of low-value samples. The samples generated by IDQL are generally positively correlated with the guidance scale, but this method will not generate any samples outside of the dataset. In other words, IDQL is unable to generate unseen, potentially high-value samples based on changes in the reward function's trends.

The samples generated by qGDP-Q and qGDP-GeQ are positively correlated with the quantile, maintaining higher diversity. Moreover, when inputting values greater than 1., they can generate samples of higher value based on the reward function's changing trends.

To better observe the impact of the input quantiles and the specified guidance scale on the generation results of qGDP-Q and qGDP-GeQ, we retrained them on the dataset "Increasing" using different quantiles $\{0.5, 0.7, 1.0, 1.3\}$ and different guidance scales $\{0.3, 1.0, 3.\}$. The generation results are shown in Figure 3.

From the distribution of generated samples, it can be observed that a guidance scale greater than 1. further guides the diffusion model to produce higher-value samples, while a guidance scale less than 1. further directs the diffusion model to generate lower-value samples. However, from another perspective, a smaller guidance scale leads to generated samples that are closer to the original data distribution and cover a larger region. For practical offline RL problems, since there is often bias in the evaluation of sample values, a larger coverage range implies greater algorithm robustness.

## 6 EXPERIMENTS

### 6.1 OFFLINE RL RESULTS

Table 1: The performance of qGDP-Q, qGDP-GeQ, and other diffusion-based algorithms on D4RL tasks. qGDP-Q, qGDP-GeQ, and DQL achieved the highest performance on the locomotion-v2 task, with qGDP-Q demonstrating performance second only to IDQL on the antmaze-v0 task. To sum up, qGDP-Q demonstrates the most superior performance among these algorithms.

| Dataset | Diffuser | DQL | SfBC | IDQL | qGDP-Q | qGDP-GeQ |
|---|---|---|---|---|---|---|
| halfcheetah-medium-v2 | 42.8 | 51.1 | 45.9 | 51.0 | **54.8±0.4** | 53.9±0.3 |
| hopper-medium-v2 | 74.3 | **90.5** | 57.1 | 65.4 | 85.7±9.2 | 78.7±16.1 |
| walker2d-medium-v2 | 79.6 | 87.0 | 77.9 | 82.5 | **92.2±1.7** | 93.1±1.4 |
| halfcheetah-medium-replay-v2 | 37.7 | **47.8** | 37.1 | 45.9 | 47.6±0.2 | 47.6±0.3 |
| hopper-medium-replay-v2 | 93.6 | **101.3** | 86.2 | 92.1 | 97.8±3.6 | 98.2±4.2 |
| walker2d-medium-replay-v2 | 70.6 | **95.5** | 65.1 | 85.1 | 96.1±2.5 | 94.1±4.6 |
| halfcheetah-medium-expert-v2 | 88.9 | **96.8** | 92.6 | 95.9 | 95.3±0.5 | 95.7±0.6 |
| hopper-medium-expert-v2 | 103.3 | 111.1 | 108.6 | 108.6 | 111.1±1.7 | **112.0±0.5** |
| walker2d-medium-expert-v2 | 106.9 | 110.1 | 109.8 | 112.7 | 114.5±1.5 | **115.9±1.2** |
| locomotion-v2 total | 697.7 | **791.2** | 680.3 | 739.2 | **795.2** | **789.1** |
| antmaze-umaze-v0 | - | 93.4 | 92.0 | 94.0 | **99.4±0.5** | 98.6±0.5 |
| antmaze-umaze-diverse-v0 | - | 66.2 | **85.3** | 80.2 | 80.4±2.6 | 79.0±3.7 |
| antmaze-medium-play-v0 | - | 76.6 | 81.3 | **84.5** | 67.8±27.5 | 69.0±27.1 |
| antmaze-medium-diverse-v0 | - | 78.6 | 82.0 | 84.8 | **89.5±3.8** | 79.2±12.0 |
| antmaze-large-play-v0 | - | 46.4 | 59.3 | **63.5** | 51.0±16.4 | 54.8±8.4 |
| antmaze-large-diverse-v0 | - | 56.6 | 45.5 | **67.9** | 64.6±4.9 | 63.0±3.3 |
| antmaze-v0 total | - | 417.8 | 445.4 | **474.6** | 452.7 | 443.6 |
| total | - | 1209.0 | 1125.7 | 1213.8 | **1247.9** | 1232.7 |

We evaluate the performance of qGDP using the domains of tasks, Locomotion and Antmaze, in D4RL Fu et al. (2020). Gym-MuJoCo and Antmaze contain multiple datasets, each containing trajectories with different quality. Rewards in Gym-MuJoCo are dense, while rewards in Antmaze are very sparse, making the latter more challenging.

In the algorithm implementation of qGDP, the quantiles for IQL are set to 0.7 and 0.9 for the Locomotion and Antmaze datasets, respectively. During the resampling process of the trained diffusion model, we iterate over quantiles in $\{0.5, 0.7, 0.8, 0.9, 1.0, 1.1, 1.2, 1.3, 1.5, 2.0\}$ and guidance scales in $\{0.03, 0.1, 0.3, 1.0, 3.0, 10.0, 30.0\}$ and record the best results corresponding to the best pair.

We select imitation learning methods such as 10%BC and DT (Chen et al., 2021) and offline RL methods such as AWAC Nair et al. (2020), Onestep RL Brandfonbrener et al. (2021), TD3+BC Fujimoto & Gu (2021), CQL Kumar et al. (2020), IQL Kostrikov et al. (2021), and Diffusion-QL Wang et al. (2022) as baselines for comparison experiments. The results of these baselines are extracted

Table 2: The performance of qGDP-Q, qGDP-GeQ, and other state-of-the-art offline RL algorithms on D4RL tasks. In comparison to these methods, qGDP-Q and qGDP-GeQ demonstrate a considerable improvement in performance.

| Dataset | 10%BC | DT | TD3+BC | Onestep | CQL | IQL | qGDP-Q | qGDP-GeQ |
|---|---|---|---|---|---|---|---|---|
| locomotion-v2 total | 666.2 | 672.6 | 677.7 | 684.9 | 698.5 | 692.4 | **795.2** | **789.1** |
| antmaze-v0 total | 134.2 | 112.2 | 163.8 | 125.4 | 303.6 | 378.0 | **452.7** | 443.6 |
| total | 800.4 | 784.8 | 841.5 | 810.3 | 1002.1 | 1070.4 | **1247.9** | 1232.7 |

from Wang et al. (2022). The normalized scores for qGDP and other baselines are recorded in Table 1 and Table 2.

Combining Tables 1 and 2, we observe that diffusion-based methods significantly outperform non-diffusion-based methods. Among these diffusion-based methods, qGDP-Q achieves the best performance, followed by qGDP-GeQ. On the locomotion-v2 task set, DQL, qGDP-Q, and qGDP-GeQ demonstrate the best performance. On the antmaze-v0 task set, IDQL achieves the best results, followed by qGDP-Q. qGDP performs well on both task sets, while DQL and IDQL perform well on only one of the task sets. One possible explanation for this phenomenon is that DQL fails to effectively generate high-value actions that have appeared in the dataset, while IDQL cannot effectively generate potential high-value actions that have not appeared in the dataset, whereas qGDP can exhibit both of these behavior patterns under different hyperparameters.

## 6.2 THE IMPACT OF QUANTILE AND GUIDANCE SCALE

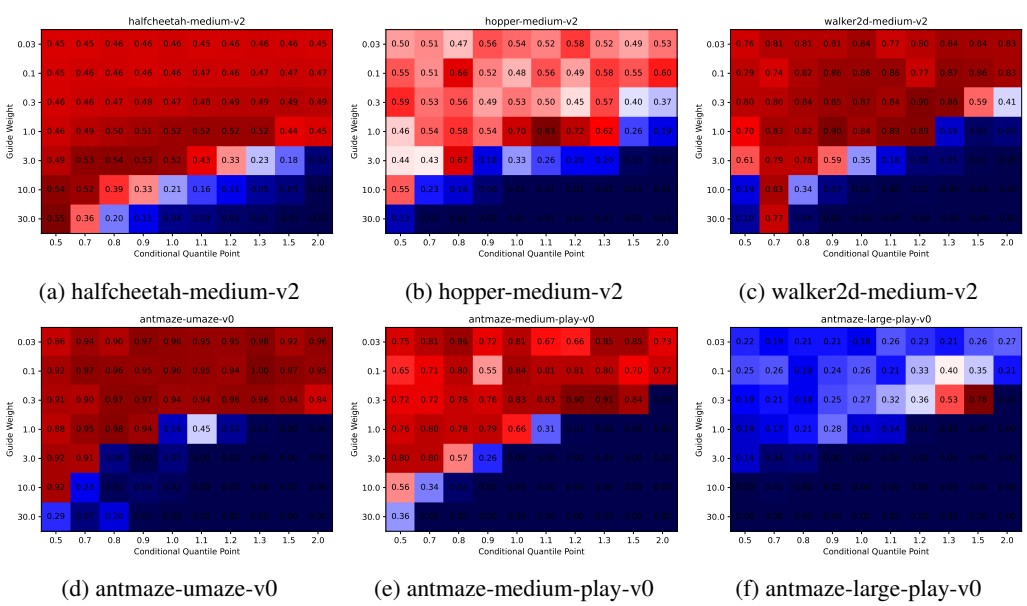

(a) halfcheetah-medium-v2    (b) hopper-medium-v2    (c) walker2d-medium-v2

(d) antmaze-umaze-v0    (e) antmaze-medium-play-v0    (f) antmaze-large-play-v0

Figure 4: The performances of qGDP-Q across multiple environments for different hyperparameters. Excessive guidance scale leads to a significant degradation in performance. In comparison to the task in antmaze-v0 (second row), the optimal for the locomotion-v2 task (first row) outcomes tend to manifest more frequently at larger guidance scales.

To observe the impact of different quantiles and guidance scales on qGDP performance, we present the results of qGDP-Q varying with hyperparameters across six tasks, as shown in Figure 4. The results indicate that an excessively large guidance scale leads to a significant drop in performance. Furthermore, optimal results for tasks in locomotion-v2 (first row) often occur with larger guidance scales compared to tasks in antmaze-v0 (second row). This further underscores the need for different balancing factors between behavior policy imitation and improvement in locomotion and antmaze task sets.

## 6.3 SMALLER HYPER-PARAMETER SPACE

Table 3: The performances qGDP-Q and qGDP-GeQ for different hyperparameter spaces.

| Dataset | qGDP-Q | qGDP-GeQ | qGDP-Q-s | qGDP-GeQ-s |
|---|---|---|---|---|
| locomotion-v2 total | 795.2 | 789.1 | 779.8 | 763.7 |
| antmaze-v0 total | 452.7 | 443.6 | 440.8 | 425.2 |
| total | 1247.9 | 1232.7 | 1220.6 | 1188.9 |

In order to verify whether a smaller hyperparameter space can yield good performance, we reduced the hyperparameter space to $q \in \{0.5, 0.7, 0.9, 1.0, 1.1, 1.3, 1.5\}$ and guidance scales $\lambda \in \{0.3, 1.0\}$. The comparative results are shown in Table 3. The performance in the smaller hyperparameter space does exhibit some decrease, but the decline in qGDP-Q is relatively small and still outperforms offline RL algorithms such as DQL, SfBC, and IDQL, which are based on diffusion.

## 6.4 RUNTIME COMPARISON

Table 4: Runtime Comparison of qGDP and DQL.

| method | critic | actor | sampling | total |
|---|---|---|---|---|
| DQL | 8.6ms×2e6 | 12.4ms×2e6 | 51.2s×40 | 12.9h |
| qGDP | 9.3ms×4e5 | 5.9ms×4e5 | 10.0s×70 | 2.0h |

In order to assess the computational efficiency of qGDP more intuitively, we ran qGDP and DQL on the same machine and recorded their execution frequencies and runtimes for each module, as shown in Table 4. The results indicate that our training approach requires fewer iterations, and the individual module's runtime is also shorter.

## 7 CONCLUSION

In this paper, we proposed an offline RL method named Quantile-Guided Diffusion Policy (qGDP). The method trains a quantile-guided network to label the training dataset and uses these labeled samples for model training conditional diffusion model and sampling the new samples in the form of classifier-free guidance. We constructed three simple tasks to analyze existing diffusion-based methods and qGDP. qGDP is able to modulate the preference of sample generation by adjusting the input of diffusion and setting the guidance scale without re-training the model, which will significantly reduce the cost of tuning the algorithm. The analysis results indicate that qGDP's generated results have a positive correlation with the input quantiles, exhibit sufficient diversity in generated samples, and are capable of generating potential high-value samples outside the training set based on the reward function's trend changes. Experimental results on D4RL datasets demonstrate that qGDP outperforms several baseline methods, including diffusion-based and other algorithms. Furthermore, we conducted a runtime comparison between qGDP and DQL to assess the computational efficiency of qGDP. The results revealed that qGDP requires fewer iterations and shorter runtimes for each module.

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

# A   ALGORITHM OVERVIEW

---
**Algorithm 1:** Quantile-Guided Diffusion
---
Initialize the diffusion model $\epsilon_\theta$, the q-value network $Q_\phi$, the target q-value network $Q_{\hat{\phi}}$, the state value network $V_\varphi$, and the quantile network $V_\eta$;

// Training Critic

**for** $iter \leftarrow 1$ *to* $N_{critic}$ **do**

    Sample B samples $(s, a, r, s')$ from training dataset $\mathcal{D}$ ;

    // Implicit Q-learning

    update $\phi \leftarrow \phi - \nabla_\phi \big(r(s, a) + V_\varphi(s') - Q_\phi(s, a)\big)^2$ ;

    update $\varphi \leftarrow \varphi - \nabla_\varphi L_2^{\tau_c}\big(Q_{\hat{\phi}}(s, a) - V_\varphi(s)\big)$ ;

    update $\hat{\phi} \leftarrow (1 - \delta)\hat{\phi} + \delta\phi$ ;

    // Quantile Learning

    update $\eta \leftarrow \eta - \nabla_\eta L^{\tau_i}\big(Q_{\hat{\phi}}(s, a) - V_\eta^i(s)\big)$

**end**

// Training Actor

**for** $iter \leftarrow 1$ *to* $N_{actor}$ **do**

    Sample $B$ samples $(s, a)$ from training dataset $\mathcal{D}$ ;

    Sample $B$ Gaussian noises $\epsilon$ from $N(\mathbf{0}; \mathbf{I})$ and B times t from $U(0, T)$ ;

    Set the samples' label through calculating (12) or sampling from (13) ;

    update $\theta \leftarrow \theta - \nabla_\theta \|\epsilon - \epsilon_\theta(\sqrt{\bar{\alpha}_t}a + \sqrt{1 - \bar{\alpha}_t}\epsilon, s, y, t)\|^2$

**end**

---

# B   DEMO EXPERIMENT DETAILS

**Dataset**   Each sample in the datasets is generated based on the following process: first, a hidden category $i \in \{1, 2, \ldots, 10\}$ is randomly sampled to determine the implicit category of the sample. Then, the sample's two-dimensional features are determined by random sampling: $(rd_i \cos(\theta) + 0.2\epsilon_x, rd_i \sin(\theta) + 0.2\epsilon_y)$, where $rd_i = \frac{i}{2}$, $\theta \sim U(0, 2\pi)$, and $\epsilon_x, \epsilon_y \sim \mathcal{N}(0, 1)$. Finally, the reward corresponding to the sample is determined by random sampling: $reward = r_i + 0.2\epsilon$, where $\epsilon_x, \epsilon_y \sim \mathcal{N}(0, 1)$. The reward sets for the three random sets are $\{1, 2, \ldots, 10\}$, $\{10, 9, \ldots, 1\}$, and $\{1, 3, \ldots, 9, 9, \ldots, 3, 1\}$.

# C   IMPLEMENTATION DETAILS

## C.1   ARCHITECTURE

**Diffusion Policy**   Our policy is an MLP-based conditional diffusion model $\epsilon_\theta(a^i|s, y, i)$, where $s$ is the state, $y$ is the input quantile, $i$ is the timestep. $\epsilon_\theta(a^i|s, y, i)$ is a three-layer fully connected network with a hidden layer width of 256, using Mish as the activation function. The timestep $i$ and the quantile $y$ are both encoded using sinusoidal positional embedding.

**Networks for Implicit Q-learning**   Q-network is a three-layer fully connected network with a hidden layer width of 256, using Mish as the activation function. Value network is a three-layer fully connected network with a hidden layer width of 256, using Mish as the activation function.

**Networks for Quantile Learning**   Value network is a three-layer fully connected network with a hidden layer width of 256, using Mish as the activation function.

## C.2   HYPERPARAMETERS

The hyperparameters of qGDP is shown in Table 5.

Table 5: Hyperparameters

| | |
|---|---|
| $\tau$ for Implicit Q-learning | 0.7 (locomotion), 0.9 (antmaze) |
| Number of quantile points $N$ | 11 (qGDP-Q), 101 (qGDP-GeQ) |
| LR (For all networks) | 3e-4 |
| Batch Size For Implicit Q-learning | 2560 |
| Batch Size For Quantile Learning | 2560 |
| Batch Size For Policy Learning | 2560 |
| Actor Grad Steps | 4e5 |
| Critic Grad Steps | 4e5 |
| Target Critic EMA $\delta$ | 0.005 |
| T | 5 |
| Beta schedule | Variance Preserving (Song et al., 2020) |
| Optimizer | Adam (Kingma & Ba, 2014) |

