# OpenReview forum: "Adjustable Quantile-Guided Diffusion Policy for Diverse Behavior Generation in Offline RL"
_ICLR.cc/2024/Conference — Submitted to ICLR 2024_

### Official Review · Reviewer_Rvvy · 2023-10-24

**Soundness:** 3 good
**Presentation:** 4 excellent
**Contribution:** 3 good
**Rating:** 5
**Confidence:** 4

**Summary:**

The paper proposes to train a diffusion policy from offline data conditional on the output of a pretrained quantile network. This allows actions to be naturally biased to be higher quality without the need to take gradients through a Q-network at test-time. The resulting approach is validated on the D4RL benchmark, with comparable performance with the prior SOTA.

**Strengths:**

- The problem setting is well-motivated and a natural extension of prior work.
- The paper enables a classifier-free method to increase the quality of actions sampled at test-time. This significantly improves inference time over the baseline.
- Clear presentation of the algorithm and informative toy experiments.

**Weaknesses:**

- **(Unclear empirical benefit)** Whilst the paper shows a strong improvement on speed, the empirical benefit of the approach is unclear, as there is only a slight gain over the baseline DQL. This is compounded by the fact that the main empirical evaluation in Section 6.1 maximizes over an extremely large 70 different hyperparameter configurations, which is extremely unrealistic for offline reinforcement learning. Coupled with the sweep in Figure 4, the main empirical evaluation is likely maximizing over statistical noise. The results over a smaller hyperparameter space in Section 6.3 do not show a clear improvement.
- **(Quantitative metric of success in toy experiments)** Whilst the authors helpfully compare the effect of different guidance and sampling schemes on their toy dataset in Figure 2-3, it is unclear what the optimal or desired behavior is. It would be helpful to include some indication of what the optimal behavior should be and some kind of metric to assess this (e.g. conforming to some desired distribution).
- **(Incomplete speed discussion)** Whilst Table 4 shows an improvement over DQL, it would be also valuable to compare runtime to related approaches, IDQL and SfBC. Furthermore, details are missing on which environment exactly is being timed in the table.

Minor:
- Figure 1 is presented without context, which is confusing for the readers. The authors should include the purpose of these datasets in the description.
- Axis labels and text in Figure 2,3,4 are very small and hard to read.
- Overall summary totals in Tables 1, 2, 3 should also have standard deviation. Given the mild improvement over the baseline, it would also be valuable to perform RLiable [1] analysis to assess the statistical significance of the method.

[1] Deep RL at the Edge of the Statistical Precipice. Agarwal et al. NeurIPS, 2021.

**Questions:**

I would greatly appreciate responses and rebuttals to the concerns raised in the weaknesses section.

---

> ### Author Response · Authors · 2023-11-18
>
> Thanks for your feedback, and we would like to provide the following clarifications.
>
> Q: (Unclear empirical benefit) Whilst the paper shows a strong improvement on speed, the empirical benefit of the approach is unclear, as there is only a slight gain over the baseline DQL. This is compounded by the fact that the main empirical evaluation in Section 6.1 maximizes over an extremely large 70 different hyperparameter configurations, which is extremely unrealistic for offline reinforcement learning. Coupled with the sweep in Figure 4, the main empirical evaluation is likely maximizing over statistical noise. The results over a smaller hyperparameter space in Section 6.3 do not show a clear improvement.
>
> A: Thank you for your suggestions.
> IDQL's experimental results are quoted from the result of IDQL-A. "-A" is refered as reported results that allow any amount of tuning and "-1" as results that only allow one hyperparameter to be tuned between domains[1]. And IDQL-1's results are only 1150.9.
> So, the performance of qGDP, 1220.6, in a smaller hyperparameter space is sufficiently good.
>
> [1] Hansen-Estruch P, Kostrikov I, Janner M, et al. Idql: Implicit q-learning as an actor-critic method with diffusion policies. arXiv preprint arXiv:2304.10573, 2023.
>
> Q: (Quantitative metric of success in toy experiments) Whilst the authors helpfully compare the effect of different guidance and sampling schemes on their toy dataset in Figure 2-3, it is unclear what the optimal or desired behavior is. It would be helpful to include some indication of what the optimal behavior should be and some kind of metric to assess this (e.g. conforming to some desired distribution).
>
> A: Thank you for your suggestions.
> The results in Fig. 2 are generated based on the toy datasets, as illustrated in Fig. 1. This experiment is primarily conducted for a qualitative analysis of the impact of hyperparameters on the results produced by different methods. The goal is to assess whether these methods exhibit generalization capabilities and to examine the effects of hyperparameters on the generated results, so there is no specific desired distribution.
>
> Q: (Incomplete speed discussion) Whilst Table 4 shows an improvement over DQL, it would be also valuable to compare runtime to related approaches, IDQL and SfBC. Furthermore, details are missing on which environment exactly is being timed in the table.
>
> A: Thank you for your suggestions.We test the release codes of DQL, IDQL, and SfB with the halfcheetah-medium-v2 task on a single RXT 2080Ti graphics card, and their runtimes are compared below:
>
> method  | critic                             | actor | sampling | total
>
> DQL        | 8.6ms$\\times$2e6     | 12.4ms$\\times$2e6 | 51.2s$\\times$40 | 12.9h
>
> IDQL      |               3.3ms$\\times$1.5e6 (critic and actor) | 70.0$\\times$6 | 1.5h
>
> SfBC      | 68.4ms$\\times$4.9e4 | 39.1ms$\\times$5.9e5 | 50.6s$\\times$20 | 7.6h
>
> qGDP     | 9.3ms$\\times$4e5      | 5.9ms$\\times$4e5 | 10.0s$\\times$70 | 2.0h
>
> where, 'critic,' 'actor,' and 'sampling' respectively refer to the time spent on value function training, policy function training, and interacting with the environment of these algorithms. The recorded form of these terms is the time required for a single update of the value function multiplied by the number of updates, the time required for a single update of the policy function multiplied by the number of updates, and the time required for model evaluation (10 interactions with the environment) multiplied by the number of evaluations.
>
> It should be noted that IDQL is implemented based on JAX, resulting in shorter runtimes. In comparison, both qGDP and DQL are implemented in PyTorch. Specifically, in the actor update phase (related to training the diffusion model), qGDP exhibits significantly shorter runtime compared to DQL. During the model evaluation phase, qGDP reduces time consumption through parallelization.
>
> Most importantly, the presented runtimes for DQL, IDQL, and SfBC are for a single run. If hyperparameter tuning is required, they would incur several times more runtime. In contrast, qGDP has completed hyperparameter tuning during the 70 model evaluation runs.

---

> > ### Comment · Reviewer_Rvvy · 2023-11-22
> > **Thanks for the repsonse**
> >
> > Thank you for the response, the clarifications on the runtime are useful for evaluating your algorithm. I will maintain my currrent score.

---

### Official Review · Reviewer_1jYd · 2023-10-28

**Soundness:** 2 fair
**Presentation:** 3 good
**Contribution:** 2 fair
**Rating:** 3
**Confidence:** 3

**Summary:**

The paper proposes a modification of IDQL with an additional quantile network for offline RL setting. The quantile network predicts values at different quintiles at the same time, thus avoiding re-training when selecting the optimal quantile value. The experiments are evaluated on two domains of D4RL tasks and show superior performance over IQL, IDQL, DQL, etc. I think the proposed method is a quite straightforward generalization of IDQL and the only novelty is the quantile network, which provides some convenience for hyperparameter searching over the quantile value. Some statements need to be further justified.

**Strengths:**

The writing is clear and the method is described well.

The experiments on bandit and D4RL are thorough, demonstrating the difference of the proposed methods over the previous ones.

The performance of the proposed algorithms beats previous SOTA results, but mostly by small margins.

**Weaknesses:**

One of my major concerns is as follows. One main advantage of the proposed method is that it can extrapolate high-value samples outside the training distributions based on the reward function’s trend. However, why is this valid for offline RL settings? The values of out-of-distribution samples are usually penalized as the pessimism of value estimation in order to achieve conservative and best performant policies in online evaluation. Please justify why simple extrapolation based on the trends of reward function is valid. One can easily construct a counterexample to make qGPD totally failed in the bandit example, by setting a first increasing then suddenly decreasing reward structure, e.g., by letting the purple regions in Fig. 3 (c)(f) to have very low reward values.

Another problem bothers me is the performance improvement over IDQL. From what I understand, the qGDP only improves over IDQL by predicting over more quantiles, so why the performance of qGDP can be better than IDQL if the quantiles considered for two methods are the same, or is the improvement just caused by qGDP searches over a larger space of quantiles or with a smaller grid size? If the improvement mainly comes from this, I cannot be convinced that qGDP is a very novel method.

Some minor suggestions:

Please illustrate more clearly about the output structure of the quantile network and the choice of N.

Please indicate what is the hyperparameter (quantile or guidance) in the caption of Fig. 2
, also provide ground truth distributions in Fig. 2.

**Questions:**

In section 4, what does the equation $V_\eta^i(s)i=1^N$ mean? I don’t get this one.

In Eq. (11), why is the quantile loss different from Eq. (8)?

In Sec. 6.4, why is qGDP faster than DQL? Please provide more explanations.

---

> ### Author Response · Authors · 2023-11-18
>
> Thanks for your feedback, and we would like to provide the following clarifications.
>
> Q: One of my major concerns is as follows. One main advantage of the proposed method is that it can extrapolate high-value samples outside the training distributions based on the reward function’s trend. However, why is this valid for offline RL settings? The values of out-of-distribution samples are usually penalized as the pessimism of value estimation in order to achieve conservative and best performant policies in online evaluation. Please justify why simple extrapolation based on the trends of reward function is valid. One can easily construct a counterexample to make qGPD totally failed in the bandit example, by setting a first increasing then suddenly decreasing reward structure, e.g., by letting the purple regions in Fig. 3 (c)(f) to have very low reward values.
>
> A: Thank you for your question.
> In practical applications of offline RL [1][2], it is typically allowed to interact with the environment to a limited extent. In these scenarios, we can utilize these interactions to select appropriate input quantile. In settings where ”a reward structure first increases and then suddenly decreases”, we can also adjust the input quantiles to a suitable value through a small number of interactions to avoid generating samples with low rewards. If applying qGDP to a scenario where online interaction is impractical, we can directly constrain the input quantiles to be less than 1.
>
> [1] Fang, Xing, et al. Offline Reinforcement Learning for Autonomous Driving with Real World Driving Data. IEEE 25th International Conference on Intelligent Transportation Systems (ITSC). IEEE, 2022.
>
> [2] Singh, Avi & Yu, Albert & Yang, Jonathan & Zhang, Jesse & Kumar, Aviral & Levine, Sergey. COG: Connecting New Skills to Past Experience with Offline Reinforcement Learning. 4th Conference on Robot Learning (CoRL 2020).
>
> Q: Another problem bothers me is the performance improvement over IDQL. From what I understand, the qGDP only improves over IDQL by predicting over more quantiles, so why the performance of qGDP can be better than IDQL if the quantiles considered for two methods are the same, or is the improvement just caused by qGDP searches over a larger space of quantiles or with a smaller grid size? If the improvement mainly comes from this, I cannot be convinced that qGDP is a very novel method.
>
> A: Thank you for your question.
> qGDP has two main advantages over IDQL: Firstly, qGDP has generalization capabilities and can generate samples with quantiles greater than 1, while IDQL cannot operate when the quantile is greater than 1. As explained in Section 5, the samples generated by qGDP align with the trend of action values (or rewards), and Section 6.2 demonstrated that qGDP performs better in some tasks when the input quantile is greater than 1. Secondly, adjusting the generation preference in IDQL requires retraining the model, whereas qGDP only requires changing the model's input. This significantly reduces the computational cost required for parameter tuning.

---

> > ### Author Response · Authors · 2023-11-18
> >
> > Q: Please illustrate more clearly about the output structure of the quantile network and the choice of N.
> >
> > A: Thank you for your suggestions.
> > The quantile network is a 4-layer fully connected neural network with the width of 256.
> > N=101
> >
> > Q: Please indicate what is the hyperparameter (quantile or guidance) in the caption of Fig. 2 , also provide ground truth distributions in Fig. 2.
> >
> > A: Thank you for your suggestions.
> > For Diffuser, this hyperparameter represents the weight $\lambda$ of the gradient guidance term in the Guided Diffusion generation process $a^i∼N(μ+λ∇J(μ),Σ)$.
> >
> > For DQL, this hyperparameter represents the guidance weight $\lambda$ when using the Q-network to guide the Diffusion model update $\arg⁡min⁡ L_{diff} (θ)-\lambda E_{a^0∼diff(θ)} Q(s,a^0) $.
> >
> > For IDQL, this hyperparameter represents the reciprocal of the temperature coefficient $\lambda$ in the sample-weighted term $exp(λ⋅Advantage)$.
> >
> > For qGDP, this hyperparameter represents the input quantile in the quantile-guided diffusion generation process.
> > The results in Fig. 2 are generated based on the toy datasets, as illustrated in Fig. 1. This experiment is primarily conducted for a qualitative analysis of the impact of hyperparameters on the results produced by different methods. The goal is to assess whether these methods exhibit generalization capabilities and to examine the effects of hyperparameters on the generated results, so there is no explicit ground truth distribution in this experiment.
> >
> > Q: In section 4, what does the equation mean? I don’t get this one.
> >
> > A: I’m sorry. This is a writing error, it should be $V_{\eta}^i (s)$.
> >
> > Q: In Eq. (11), why is the quantile loss different from Eq. (8)?
> >
> > A: Thank you for your question.
> > The form of Eq. (8) is primarily used for assessing action values, while the form in Eq. (11) is employed to obtain values corresponding to different quantiles. Under the form of Eq. (11), the distribution of values corresponding to quantiles is more uniform.
> >
> > Q: In Sec. 6.4, why is qGDP faster than DQL? Please provide more explanations.
> >
> > A: Thank you for your question.
> > When DQL uses the Q-network to guide the training of the diffusion model, it treats the generation process of the Diffusion model as a policy network of size d*N and optimizes it through gradient backpropagation, where d is the depth of the neural network in the diffusion model, and N is the number of diffusion timesteps. In other words, DQL involves computing gradients for a relatively deep neural network. qGDP is designed to train the diffusion model to generate samples at different quantiles. Therefore, the computational cost of qGDP is comparable to that of a regular Diffusion model.

---

> > > ### Comment · Reviewer_1jYd · 2023-11-22
> > >
> > > Thanks for the response.
> > >
> > > First, the paper can be modified during the rebuttal phase.
> > >
> > > Offline training with online interaction will be a very different setting than purely offline training, and it is not verified in the paper. So the answer is not valid for the effectiveness of extrapolation with only offline training.
> > >
> > > > ground truth in Fig. 2
> > >
> > > I indicate the ground truth by dataset distribution.
> > >
> > > > In Eq. (11), why is the quantile loss different from Eq. (8)?
> > >
> > > I still don't understand why the loss form in terms of $u$ has to be different.
> > >
> > > > In other words, DQL involves computing gradients for a relatively deep neural network.
> > >
> > > Does it indicate the qGDP is using a smaller network? If not, why is it faster, both qGDP and DQL are diffusion policies.
> > >
> > > Given above unsolved questions, I will not the change the score.

---

### Official Review · Reviewer_nVWg · 2023-10-31

**Soundness:** 2 fair
**Presentation:** 2 fair
**Contribution:** 2 fair
**Rating:** 3
**Confidence:** 4

**Summary:**

This paper proposes a novel offline RL algorithm qGDP using quantile labeling and conditional diffusion models. It shows competitive results on D4RL benchmarks compared to other diffusion-based methods. The approach allows tuning action distributions without retraining. However, there are some limitations in justifying the quantile conditioning and comparing against other offline RL methods.

**Strengths:**

1. Achieves state-of-the-art results among diffusion-based offline RL methods on D4RL.
2. Novel approach of conditioning diffusion on quantile labels for offline RL.
3. Allows flexible tuning of action distributions without retraining.

**Weaknesses:**

1. The quantile labels rely on the action values Q learned by IQL, which may have overestimation bias. Errors in Q would propagate to incorrect quantile labels, which could be amplified for higher quantiles. This could negatively impact the quality of the diffusion model training.
2. The quantile input y for guiding the diffusion model is constrained to the range [0,1] or not ?  It will limit the scope of behavioral patterns that can be generated. Values greater than 1 may allow more generalization, but this is not explored. And I expected to find the experiment results of different value of y for different generation, but failed.
3. While the quantile input is motivated by IDQL, the paper does not sufficiently differentiate the advantages of the quantile mechanism compared to IDQL itself. The paper does not sufficiently differentiate the advantages of the quantile input compared to just using weighted actions like in IDQL. The results between qGDP and IDQL in Table 1 are fairly close. More analysis is needed to clearly explain the relationship between the two methods and highlight the unique benefits of the quantile inputs.
4. The paper over-emphasizes the impact of tuning the guidance scale, even though adjusting the scale for diversity is intrinsic to diffusion models themselves. However, the main innovation of the paper is using the quantile network to label samples and train the diffusion policy conditioned on quantiles. More analysis is needed on how the quantile inputs specifically affect diversity, rather than just tuning the guidance scale which is a general feature of diffusion models.
5. In Table 4, the runtime of the proposed offline RL algorithm qGDP is compared to the older online RL algorithm DQL? I cannot get the meaning of this comparison.  For a fair runtime comparison, it would make more sense to compare against other recent offline RL algorithms such as IDQL.

**Questions:**

Please see the above weakness and address my concerns.

---

> ### Author Response · Authors · 2023-11-18
>
> Thanks for your feedback, and we would like to provide the following clarifications.
>
> Q: The quantile labels rely on the action values Q learned by IQL, which may have overestimation bias. Errors in Q would propagate to incorrect quantile labels, which could be amplified for higher quantiles. This could negatively impact the quality of the diffusion model training.
>
> A: Thank you for your question.
> In our code implementation, we have utilized the Clipped Double Q-learning technique provided by TD3[1] to mitigate Overestimation Bias.
> [1] Fujimoto S, Hoof H, Meger D. Addressing function approximation error in actor-critic methods[C]//International conference on machine learning. PMLR, 2018: 1587-1596.
>
> Q: The quantile input y for guiding the diffusion model is constrained to the range [0,1] or not ? It will limit the scope of behavioral patterns that can be generated. Values greater than 1 may allow more generalization, but this is not explored. And I expected to find the experiment results of different value of y for different generation, but failed.
> The paper over-emphasizes the impact of tuning the guidance scale, even though adjusting the scale for diversity is intrinsic to diffusion models themselves. However, the main innovation of the paper is using the quantile network to label samples and train the diffusion policy conditioned on quantiles. More analysis is needed on how the quantile inputs specifically affect diversity, rather than just tuning the guidance scale which is a general feature of diffusion models.
>
> A: Thank you for your question.
> The quantile input can take values larger than 1. In Figure 3, we visualize the generation results of qGQP under different guidance scales and quantile inputs. In each subfigure, points of different colors represent generated samples for quantile inputs y=0.5, 0.7, 1.0, and 1.3. The figure is evident that qGQP exhibits strong generalization capabilities, generating samples not present in the dataset and consistent with the underlying trend.
> Furthermore, on the D4RL dataset, we experimented with different quantile inputs. The results indicate that both quantile input and guidance scale significantly impact the algorithm's performance, as shown in Figure 4.
>
> Q: While the quantile input is motivated by IDQL, the paper does not sufficiently differentiate the advantages of the quantile mechanism compared to IDQL itself. The paper does not sufficiently differentiate the advantages of the quantile input compared to just using weighted actions like in IDQL. The results between qGDP and IDQL in Table 1 are fairly close. More analysis is needed to clearly explain the relationship between the two methods and highlight the unique benefits of the quantile inputs.
>
> A: Thank you for your suggestions.
> qGDP has two main advantages over IDQL: Firstly, IDQL increases the probability of the diffusion model generating high-value actions by reweighting samples. So it cannot generate samples with higher values outside the dataset. In contrast, qGDP learns the probability distribution of actions conditioned on different quantiles. Therefore, when the true action value function changes smoothly, qGDP can output action distributions corresponding to higher action values (associated with quantile inputs greater than 1). Secondly, adjusting the generation preference in IDQL requires retraining the model, whereas qGDP only requires changing the model's input. This significantly reduces the computational cost required for parameter tuning.
> On the locomotion-v2 task set and the tasks of antmaze-umaze-v0 and antmaze-umaze-diverse-v0, our method significantly outperforms IDQL. In the remaining tasks, the reason for qGDP being weaker than IDQL may be attributed to the excessively large state space. The probability of each state being labeled with different labels significantly decreases, leading to insufficient training for quantile-guided diffusion.

---

> > ### Author Response · Authors · 2023-11-18
> >
> > Q: In Table 4, the runtime of the proposed offline RL algorithm qGDP is compared to the older online RL algorithm DQL? I cannot get the meaning of this comparison. For a fair runtime comparison, it would make more sense to compare against other recent offline RL algorithms such as IDQL.
> >
> > A: Thank you for your suggestions.
> > We test the release codes of DQL, IDQL, and SfB with the halfcheetah-medium-v2 task on a single RXT 2080Ti graphics card, and their runtimes are compared below:
> >
> > method  | critic                             | actor | sampling | total
> >
> > DQL        | 8.6ms$\\times$2e6     | 12.4ms$\\times$2e6 | 51.2s$\\times$40 | 12.9h
> >
> > IDQL      |               3.3ms$\\times$1.5e6 (critic and actor) | 70.0$\\times$6 | 1.5h
> >
> > SfBC      | 68.4ms$\\times$4.9e4 | 39.1ms$\\times$5.9e5 | 50.6s$\\times$20 | 7.6h
> >
> > qGDP     | 9.3ms$\\times$4e5      | 5.9ms$\\times$4e5 | 10.0s$\\times$70 | 2.0h
> >
> > where, 'critic,' 'actor,' and 'sampling' respectively refer to the time spent on value function training, policy function training, and interacting with the environment of these algorithms. The recorded form of these terms is the time required for a single update of the value function multiplied by the number of updates, the time required for a single update of the policy function multiplied by the number of updates, and the time required for model evaluation (10 interactions with the environment) multiplied by the number of evaluations.
> >
> > It should be noted that IDQL is implemented based on JAX, resulting in shorter runtimes. In comparison, both qGDP and DQL are implemented in PyTorch. Specifically, in the actor update phase (related to training the diffusion model), qGDP exhibits significantly shorter runtime compared to DQL. During the model evaluation phase, qGDP reduces time consumption through parallelization.
> >
> > Most importantly, the presented runtimes for DQL, IDQL, and SfBC are for a single run. If hyperparameter tuning is required, they would incur several times more runtime. In contrast, qGDP has completed hyperparameter tuning during the 70 model evaluation runs.

---

> > > ### Comment · Reviewer_nVWg · 2023-11-23
> > > **Thanks for the response.**
> > >
> > > Thanks much for your response!
> > > 1. You make fair points around qGDP's advantages in generalization and computational efficiency over IDQL. However, the performance between qGDP and IDQL on many tasks is still fairly close. I would like to see deeper analysis explaining why qGDP lags behind IDQL on some tasks and more clearly differentiating the unique benefits of the quantile mechanism.
> > > 2.  The diversity primarily stems from interpolating between conditioned policy generations, rather than exhibiting more fundamentally novel behaviors. The novelty seems somewhat incrementally limited.
> > > Overall, while the rebuttal helps clarify some points, I believe my original major concerns around understanding the impact of the quantile inputs and comparing to IDQL still remain. As such, I would like to maintain my original review score.

---

### Official Review · Reviewer_eP2Y · 2023-11-02

**Soundness:** 3 good
**Presentation:** 3 good
**Contribution:** 2 fair
**Rating:** 5
**Confidence:** 3

**Summary:**

The paper presents a novel offline Reinforcement Learning (RL) algorithm, Quantile-Guided Diffusion Policy (qGDP), which aims to learn optimal policies from pre-collected data without the need for costly or impractical online interactions. The qGDP method involves a quantile network for dataset labeling, and leverages labeled data to train a diffusion model, enabling sample generation for policy improvement or imitation. The flexibility of the qGDP is highlighted in its ability to modify action generation preferences without retraining, promising computational efficiency. Experimental validation is provided through the D4RL benchmark, where qGDP shows superior performance and efficiency relative to existing diffusion-based approaches.

**Strengths:**

1. The manuscript is well-structured, providing clear insight into the proposed method and its implications for offline RL.

2. It addresses an essential challenge in RL regarding behavior generation diversity, which is critical for robust policy learning.

3. Section 5 effectively elucidates the distinct advantages of qGDP over other methods, offering valuable context and justification for the proposed approach.

**Weaknesses:**

1. The paper lacks an introductory explanation of quantile networks and their relevance to the proposed method, potentially hindering comprehension for readers less versed in the domain. The authors are encouraged to elaborate on the concept and role of quantile networks within qGDP to provide readers with a foundational understanding of the methodology. An illustration to overview the entire work would be helpful.

2. The core innovation, applying quantile labels to guide the diffusion process, appears somewhat incremental, casting doubt on the method's novelty. Clarification on the specific novelties and contributions of qGDP  would be beneficial.

3. The introduction does not adequately articulate how qGDP surmounts the limitations of prior diffusion-based offline RL methods. It would be advantageous to outline explicitly how qGDP addresses the deficits of previous diffusion-based methods in the introduction.

4. While informative, Section 5 may benefit from condensation to improve readability and maintain focus. Considering the length of Section 5, it is recommended to distill the content to the most salient points to maintain the reader's engagement and enhance clarity.

**Questions:**

Please refer to the weaknesses.

---

> ### Author Response · Authors · 2023-11-18
>
> Thanks for your feedback, and we would like to provide the following clarifications.
>
> Q: The paper lacks an introductory explanation of quantile networks and their relevance to the proposed method, potentially hindering comprehension for readers less versed in the domain...
>
> A: Thank you for your suggestions. The relevant explanations are provided below, and we will incorporate these details into the main text in the subsequent revisions：
> The quantile network, proposed by Dabney et al.[1], is designed to describe the action value distribution for each state-action pair (i.e., estimating the values corresponding to different quantiles under this distribution). It constructs the corresponding Distributional Bellman Operator to improve the performance of DQN. IQL employs a similar approach to characterize the action value distribution under the behavioral policy, and uses the values corresponding to high quantile points as an approximate estimate for the optimal action. This helps in avoiding querying out-of-sample actions in the TD loss.
> In this work, we utilize the quantile network to describe the distribution of action values under the behavioral policy, assign different labels to samples based on the value according to the quantiles, and finally use these labeled samples to train a diffusion model. This trained diffusion model is able to generate appropriate actions based on the input quantiles and guidance scales and exhibits different decision preferences.
> [1] Dabney W, Rowland M, Bellemare M, et al. Distributional reinforcement learning with quantile regression, Proceedings of the AAAI Conference on Artificial Intelligence. 2018, 32(1).
>
> Q: The core innovation, applying quantile labels to guide the diffusion process, appears somewhat incremental, casting doubt on the method's novelty. Clarification on the specific novelties and contributions of qGDP would be beneficial.
>
> A: Thank you for your suggestions. The main contributions of qGPD are as follows:
> Hyperparameter tuning is the big problems that still lack a satisfying solution.[2] Most offline RL methods select the optimal hyperparameters by evaluating the performances of models trained with different hyperparameters by interacting with the environment[3]. This implies that these methods require repeating the training process much times. Our approach achieves a behavior-controllable policy with just one training iteration. By adjusting the input quantile and guidance scale, this policy can be biased towards generating actions with higher estimated values or towards actions that closely resemble the behavioral policy.
> [2] Levine S, Kumar A, Tucker G, et al. Offline reinforcement learning: Tutorial, review, and perspectives on open problems[J]. arXiv preprint arXiv:2005.01643, 2020.
> [3] Prudencio R F, Maximo M R O A, Colombini E L. A survey on offline reinforcement learning: Taxonomy, review, and open problems[J]. IEEE Transactions on Neural Networks and Learning Systems, 2023.
>
> Q: The introduction does not adequately articulate how qGDP surmounts the limitations of prior diffusion-based offline RL methods. It would be advantageous to outline explicitly how qGDP addresses the deficits of previous diffusion-based methods in the introduction.
>
> A: Thank you for your suggestions. The limitations of prior diffusion-based offline RL methods and the advantages of qGDP are listed below:
> a. IDQL increases the probability of the diffusion model generating high-value actions by reweighting samples, so it cannot generate samples with higher values outside the dataset. In contrast, qGDP learns the probability distribution of actions conditioned on different quantiles, thus qGDP can output action distributions corresponding to higher action values (associated with quantile inputs greater than 1).
> b. DQL uses the Q-network to guide the training of the diffusion model, so it involves computing gradients for a relatively deep neural network. qGDP is designed to train the diffusion model to generate samples at different quantiles, so the computational cost of qGDP is comparable to that of a regular Diffusion model.
> c. Adjusting the generation preference in other diffusion-based offline RL methods requires retraining the model, whereas qGDP only requires changing the model's input. This significantly reduces the computational cost required for parameter tuning.

---

> > ### Author Response · Authors · 2023-11-18
> >
> > Q：While informative, Section 5 may benefit from condensation to improve readability and maintain focus. Considering the length of Section 5, it is recommended to distill the content to the most salient points to maintain the reader's engagement and enhance clarity.
> >
> > A: Thank you for your suggestions. The main conclusions from Section 5 are as follows:
> > 1 Each approach can generally approximate the distribution in the dataset or maximize the value of generated samples by adjusting hyperparameters.
> > 2 For Diffuser, the highest-value samples on the Middle dataset do not correspond to the largest guidance scale
> > 3 IDQL is unable to generate unseen, potentially high-value samples based on changes in the reward function’s trends.
> > 4 For qGDP-Q and qGDP-GeQ, when inputting values greater than 1., they can generate samples of higher value based on the reward function’s changing trends.

---

> > > ### Comment · Reviewer_eP2Y · 2023-11-23
> > >
> > > Thanks much for your feedback! Some of my concerns are addressed. However, for the question of the core innovation, I am not convinced that Hyperparameter tuning is the core contribution of this work. First, while it can generate behaviors with different performance, the training of the model itself still needs tuning its hyper parameters. Second, the method introduce extra hyper-parameters, like the number of the quantiles. Third, it seems like that in your manuscript, there is not sufficient description, discussion, and empirical analysis to support this point. As a result, I will maintain my score.

---

### Meta-Review · Area_Chair_r6T6 · 2023-12-09

**Metareview:**

This paper presents an approach to condition a diffusion-model based behavior policy towards high rewarding actions by conditioning on quantiles of an estimated value function. The results on D4RL are kind of mixed (which could just be due to saturation of the benchmark), but I felt that the paper could deserve more analysis. In particular, the difference between this approach and IDQL is conditioning on quantiles vs weighting by advantages. This is equivalent to two changes -- (1) conditioning vs weighting, and (2) quantiles vs advantages. I felt that the paper didn't make a rigorous comparison of how each of these changes affects performance.

The analysis in Section 5 is interesting, but seems incomplete. The claims about hyperparameter tuning are not rigorous -- likely need to be shown and analyzed more carefully across a wide range of "stress-test" data compositions along with an rigorous explanation of why we would expect the method to be better at dealing with sensitivity. I personally do think that conditioning on quantiles can be quite nice and might have some of these benefits, but the analysis in the paper does not show this concretely.

Overall, the reviewers made some good points as well, so I would encourage the authors to carefully go over their comments and revise the paper for the next conference. I personally think that a more analysis-focused vs methods-focused paper on some of these design choices (that I outlined above) could be quite nice for the community.

**Justification For Why Not Higher Score:**

Outlined in my meta review: lack of concrete contributions and significant impact

**Justification For Why Not Lower Score:**

N/A

---

### Decision · Program_Chairs · 2024-01-16

Reject